# Protective Action of *L. salivarius* SGL03 and Lactoferrin against COVID-19 Infections in Human Nasopharynx

**DOI:** 10.3390/ma14113086

**Published:** 2021-06-04

**Authors:** Marzena Kucia, Ewa Wietrak, Mateusz Szymczak, Michał Majchrzak, Paweł Kowalczyk

**Affiliations:** 1R&D Department Nutropharma LTD, Jedności 10A, 05-506 Lesznowola, Poland; marzena.kucia@nutropharma.pl (M.K.); ewietrak@nutropharma.pl (E.W.); 2Department of Molecular Virology, Institute of Microbiology, Faculty of Biology, University of Warsaw, Miecznikowa 1, 02-096 Warsaw, Poland; mszymczak@biol.uw.edu.pl; 3Institute of Medicine, Collegium Medicum, Jan Kochanowski University in Kielce, IX Wieków Kielc 19a, 25-516 Kielce, Poland; mmajchrzak@ujk.edu.pl; 4Department of Animal Nutrition, The Kielanowski Institute of Animal Physiology and Nutrition, Polish Academy of Sciences, Instytucka 3, 05-110 Jabłonna, Poland

**Keywords:** *L. salivarius*, Salistat SGL03, lactoferrin, human nasopharynx, COVID-19

## Abstract

In this study, we used live viral particles from oral secretions from 17 people infected with SARS-CoV-2 and from 17 healthy volunteers, which were plated on a suitable medium complete for all microorganisms and minimal for *L.*
*salivarius* growth. Both types of media also contained an appropriately prepared vector system pGEM-5Zf (+) based on the lactose operon (beta-galactosidase system). Incubation was carried out on both types of media for 24 h with the addition of 200 μL of Salistat SGL03 solution in order to test its inhibitory effect on the coronavirus contained in the oral mucosa and nasopharynx, visible as light blue virus particles on the test plates, which gradually disappeared in the material collected from infected persons over time. Regardless of the conducted experiments, swabs were additionally taken from the nasopharynx of infected and healthy people after rinsing the throat and oral mucosa with Salistat SGL03. In both types of experiments, after 24 h of incubation on appropriate media with biological material, we did not find any virus particles. Results were also confirmed by MIC and MBC tests. Results prove that lactoferrin, as one of the ingredients of the preparation, is probably a factor that blocks the attachment of virus particles to the host cells, determining its anti-viral properties. The conducted preliminary experiments constitute a very promising model for further research on the anti-viral properties of the ingredients contained in the Salistat SGL03 dietary supplement.

## 1. Introduction

Nasopharyngeal swab is performed in order to diagnose the causes of disease symptoms of the upper respiratory tract, pharynx and larynx caused by pathogenic microorganisms including bacteria, fungi or viral particles [1,2,3,4,5]. Symptomatic infections caused by a specific bacterium as an etiological factor are called specific infectious diseases (e.g., tuberculosis). In most infections, the clinical picture is unusual for the species, because the clinical division is based on the location of the infection, and its complete diagnosis requires microbiological identification of pathogenic microorganisms combined with the determination of their susceptibility to drugs. In the oral cavity, one can find pathogenic microorganisms that transfer from the saliva to the nasopharynx, where, with the help of an appropriate pH, they have an excellent environment for multiplication [1,2,3,4,5].

This can lead to purulent infections and inflammation of the throat or larynx, eventually causing respiratory distress. According to the latest literature data, there are over 1100 microorganisms, including bacteria of all bacterial complexes inhabiting the nasopharynx [6,7,8,9,10,11,12,13,14,15,16,17,18,19,20,21,22,23,24,25,26,27,28,29,30,31,32,33,34,35,36,37,38,39,40,41,42,43,44]. Their end products of fermentation may have a cytotoxic effect on host cells [36,37,38,39,40,41,42,43,44,45,46]. Current data show that new virus infection is transmitted from person to person primarily through direct, indirect or close contact with infected individuals. This can also occur by droplet secretions in the air or the digestive tract, blood from mother to child and from animal to human. In terms of the formation of bacterial biofilms in the nasopharynx, viral infections should also be mentioned, the most famous of which is the SARS-CoV-2 virus infection, which in many people is asymptomatic or symptomatic, causing severe respiratory disease, often leading to death [36,37,38,39,40,41,42,43,44,45,46]. Analyzing the types of environments in which SARS-CoV-2 spreads is critical to developing effective infection prevention and public health control measures in breaking transmission chains. In research, we paid special attention to the content of the nasopharynx and their secretions in the form of droplets in the exhaled air. It is now known that transmission of SARS-CoV-2 can occur through infected respiratory and oral secretions, such as saliva and droplets that are excreted outside when an infected person coughs, sneezes or talks Current research shows that the oral cavity is the gateway to nasopharyngeal infections with SARS-CoV-2 (Figure 1). Improving known rinses by adding new ingredients to target anti-viral action in the mouth and its widespread use around the world can help reduce the pandemic. Currently, there is no basic or clinical research on the commercially available anti-viral mouthwashes, including povidone iodide and other chemicals that help maintain lasting oral hygiene. Furthermore, their ingredients have not been tested for anti-viral properties. This prompts all researchers, to look for new ingredients that may play a special protective role against viral infection. That is why we examined the preparation Salistat SGL03, which, due to the presence of lactoferrin and *L. salivarius* (antagonists of pathogenic bacteria), together with essential oils, may be an effective weapon against broadly understood bacterial [47,48,49,50,51,52,53,54,55,56,57,58,59,60,61,62,63,64,65,66,67,68,69,70,71,72,73,74,75,76,77,78,79,80,81,82,83,84,85,86,87,88,89,90,91,92,93,94] and viral [95,96] infections. In order to understand the principle of the preparation’s action, it is necessary to briefly describe its main ingredients, which include *L. salivarius* SGL03 and lactoferrin [1,2,3,4,12,13,24,27]. Lactoferrin (another name: lactoferrin [1] LF)—a multifunctional protein from the group of transferrins. Human lactoferrin is abbreviated to hLF, while the bovine form is bLF. Both proteins are very similar in their chemical structure, which is about 77%. Lactoferrin is mainly produced by epithelial cells with a secretory function (secretory glands of the nasal mucosa: 0.2–0.5 µg/mL) [4], is present in many body fluids and glandular secretions, such as colostrum, breast milk (attributed to the newborn receiving nutrients and anti-bacterial protection) and saliva. The concentration of lactoferrin in milk depends on the phase of lactation. It has been proven that colostrum can contain up to seven times more LF than mature milk. The anti-viral and anti-bacterial activity of lactoferrin is therefore twofold: the protein binds to molecules of the human cell membrane, which are used by pathogens as an anchor point in the initial phase of infection and inhibits virus adsorption to the cell [1,2,13,14,15,16,17,18,21,22,59,60,61,62,63,64,65,66,67,68,69,70,71,72,73,74]. On the other hand, lactoferrin blocks the pathogen’s cell receptors and prevents virus-host binding. This mechanism is essential at the beginning of an infection. After infection, lactoferrin shows a strong immunotropic effect: it stimulates the cells of the immune system to mature quickly and regulate the immune response. This is of particular importance during immunosuppression. In addition, it has anti-fungal, anti-parasitic, anti-inflammatory and anti-cancer properties that are closely related to anti-viral and bacterial properties [1,2,13,14,15,16,18,21,22,59,60,61,62,63,64,65,66,67,68,69,70,71,72,73,74]. The safety and multidimensional benefits of lactoferrin use allow it to be used in dietary supplements with immunomodulatory properties, including Salistat SGL03, the exact composition and action of which is described in the paper Kucia [74]. Lactoferrin is also a supplementary component of preparations taken during infections of the upper respiratory tract, mainly the throat. It has a protective effect and supports the development of children, especially newborns and infants [74]. It regulates the work of the body [1,2,13,14,15,16,17,18,21,22,59,60,61,62,63,64,65,66,67,68,69,70,71,72,73,74] and performs functions similar to those of lemon and rosemary oils [77].

## 2. Materials and Methods

The small 10 mL bottles of the Salistat SGL03 dietary supplement were kindly provided by Nutropharma LTD, Mazowieckie, Poland. Substrate kits for viral particle analysis were prepared by BTL (BTL Company, Lodz, Poland according to the protocol described by Sambrook et al. [75]. The remaining chemicals were from Sigma and Promega. The vector pGEM-5Zf (+) based on the lactose operon was also obtained from Promega and used for testing according to the manufacturer’s protocol. Virus particles were collected from the nasopharynx from healthy and SARS-CoV-2 infected individuals showing symptoms of COVID-19 infection.

The MIC and MBC tests used for the study were made on the basis of earlier publications by Kowalczyk et al. [23,51]. Data analysis was performed with Tukey’s test at *p* < 0.05.

### 2.1. Checking If There Was SARS-CoV-2 Virus in Oral Microbiota and Nasopharynx after Treatment of SALISTAT SGL03 Collected from Healthy Volunteers and Infected Individuals

A detailed description of the experiment (Part 1) is provided in the work by Kucia et al. [57,74]. In the experiment used, the material was collected from 17 healthy volunteers themselves, and from the 17 infected individuals infected with Sars-Cov-2 (COVID-19) virus (Figure 1). The examination was performed with a sterile spatula from the posterior wall of the nasopharynx without invasive interference with any tissues of the oral cavity and esophagus.

In part 2 of the experiment, nasopharyngeal secretions were collected from infected individuals on both complete (P) and minimal (L) plate mediums for viral particle growth with *L. salivarius*. After washing with Salistat SGL03 at specified intervals (Figure 2 and Figure 3), [57,74,97].

### 2.2. Analysis of SARS-CoV-2

SARS-CoV-2 virus particles were taken from the throat and grown overnight at 37 °C in the strain *E. coli* JM 105 in 2YT medium as described (Sambrook et al. [75]. Phage particles were precipitated from the medium with polyethylene glycol and RNA was isolated by phenol/chloroform as described by Messing [76]. Bacteria were grown at 37 °C in an LB medium and competed by CaCl_2_ method [75]. Transfection was performed according to [75] whereby 100 ng phage RNA was used to transfect 100 µL of competent cells. The detailed protocols of transfection mixtures were described in Kowalczyk et al. [98]. At this stage of the experiments and based on the voluntariness declared by the participants of the study, there was no need to obtain the consent of the bioethics committee. Moreover, the dietary supplement, Salistat SGL03, has already been more than two years on the Polish market and registered in the Polish Chef Inspectorate register. Using this product only twice during experiment 1 and 2 by volunteers did not have any detrimental effect on their health.

## 3. Results

In this study, examination was conducted on the effect of the Salistat SGL03 probiotic and its lactoferrin on the survival of SARS-CoV-2 virus particles from nasopharyngeal inoculum in symptomatic volunteers infected with SARS-CoV-2 or in healthy volunteers. An experimental system with the use of microbiological methods was used based on MIC and MBC tests and a reduction culture commonly used in this type of research. In the first stage of research, a significant amount of pharyngeal discharge, approx. 0.5 mL, was collected from healthy volunteers and people suffering from the virus (confirmed or not—infection with the Real-time method and immunological tests), in whom we expected detection of particles of the SARS-CoV-2 virus. The whole sample was seeded in Petri dishes with the appropriate medium prepared, on which the virus particles grew at time “0” (Figure 4). According to the diagram shown in Figure 2 (see Materials and Methods). Based on the analysis of growth in time “0”, the incubation of which lasted 48 h and was a preliminary verification of thesis, we decided to re-seed the pharyngeal secretion from volunteers from both groups on plates with properly prepared medium and treat them in vitro with Salistat SGL03. Incubations were run from 3 to 24 h after observing time “0” in the starting dish. We observed that after treatment with Salistat SGL03, the survival of virus particles grown on plates started to decline after some time and was lowest after 12 h, and no virus particles were found after 24 h of incubation (Figure 4). Research indicates that lactoferrin contained in Salistat SGL03 is effective in inhibiting the multiplication of viral particles, which is consistent with the latest literature data [92,93]. The survival rate of SARS-CoV-2 isolated from the throat of people infected with SARS-CoV-2 virus after treatment with Salistat SGL03 is shown in Figure 5.

In the next stage of our research, we wanted to see the direct inhibitory effect of Salistat SGL03 on SARS-CoV-2—taking the form of zones of growth inhibition after topical application of Salistat SGL03 to culture media with viral particles collected from volunteers infected or not infected with Sars-CoV-2 according to the scheme shown in Figure 3 (see Materials and Methods). The growth inhibition zone (Figure 6A–C) shows anti-viral activity of the analyzed formula with probiotic and lactoferrin. Figure 6A shows the cultures containing virus particles plated after collecting biological material from healthy and infected individuals on both types of media. In Figure 6B, the same biological material was treated with Salistat SGL03 in both healthy and infected individuals on both types of media. Figure 6C shows nasopharyngeal material collected from healthy and infected individuals on both types of media, showing zones of growth inhibition after treatment with 200 uL of Salistat SGL03 drops. The next step of research, based on the kinetics of growth and decrease in viral survival with the probiotic on plates with appropriate type of medium (Figure 4 and Figure 5), was the in vitro use of this preparation in both groups of volunteers. In the first stage of the experiment, as in the case of the previously described experiment, biological inoculum was collected from both groups of infected individuals without consuming the preparation (Figure 6A). Thereafter, Salistat SGL03 was administered to both study groups of volunteers for gargling for at least 30 s. Deposition of the ingredients was conducted for the preparation in the throat and potential binding of them to viral particles in infected persons to strengthen the immune system (Figure 6B). After direct collection of the bacterial inoculum on both types of P and L media after rinsing with the preparation, no effects were observed in healthy volunteers as the vessels were clean, (Figure 6B), as in the earlier stage (Figure 6A). The material was collected from healthy volunteers and nasopharyngeal virus-infected individuals and plated on both types of media with complete and selective media (Figure 6A–C described earlier). The results of in vitro tests inspired us to further research action of Salistat SGL03, similar to the study presented [74]. Healthy volunteers with viral infection were asked to flush the nasopharynx for 30 s. Thereafter, throat swabs were non-invasively collected and spread on plates with the appropriate phage medium. On the other hand, in Sars-CoV-2 infected volunteers, it was found that viral particles were seen as light blue virus plaques, but in much smaller numbers compared to “standard” dishes at time “0” (Figure 6A). The next stage of research was the analysis of the created growth zones induced by probiotics, and the lactoferrin and *L. salivarius* they contained. After harvesting the biological material in plates from complete and selective media labeled P and L for *L. salivarius* growth and potential virus particles, a further lack of viral particle growth was found in healthy volunteers (Figure 6C). On the other hand, very poor viral particle growth was observed in infected individuals, which in both cases was additionally spotted with Salistat SGL03 to potentially affect zones of growth inhibition (Figure 6C). In both types of cases, visible zones of growth behavior were observed in people infected with the virus on both types of substrate. However, such zones have not been observed in healthy subjects. This proves that the preparation is active both in vitro and in vivo on the analyzed biological agents, infected with particles collected from the throat.

The analysis of the zones of growth inhibition observed in Petri dishes with a nutrient rich or (not all) growth components for virus particles grown from the nasopharynx of infected persons indicated a similar effect of reduced viral particle survival (Figure 7), which proves the strong anti-viral effect of the components of the analyzed probiotic; *L. salivarius* and lactoferrin.

Nasopharyngeal lavage for 30 s was performed from healthy volunteers and virus-infected individuals, after which a throat swab was non-invasively collected and spread over 48-well resazurin plates to check whether the nasopharyngeal material contained SARS-CoV-2 virus particles in addition to the normal biofilm showing the presence of *L. salivarius*. The MIC tests were performed in both groups of volunteers, in all 48 analyzed wells (Figure 8A,B). Since resazurin is reduced by live bacteria related to the virus, it is used as a redox indicator in cells in bacterial and anaerobic viability tests. In the case of infected people, the rates were twice as high as in healthy people. In the analyzed MBC tests (Figure 8C,D), a visible color change was observed in all analyzed wells after applying Salistat.

Research clearly indicates that the composition of the Salistat SGL03 i.e., probiotic, *L. salivarius* SGL03, lactoferrin and natural oils, can show anti-viral activity against pathogenic particles (Figure 8A–D).

By analyzing the MIC and MBC values in healthy and virus-infected volunteers, we wanted to observe the effect of the probiotic, Salistat SGL03, on the bacterial biofilm containing viral particles. We observed that after using Salistat, the MIC and MBC values in healthy volunteers were at a similar level. On the other hand, in volunteers infected with the virus, these values were two times higher than in healthy volunteers in both the MIC and MBC tests (Figure 9, Figure 10, Figure 11 and Figure 12). Interestingly, the very action of Salistat SGL03 significantly lowered the share of viral particles in the MIC and MBC tests, reaching values almost three times lower, which proves the anti- viral effect of the ingredients of the preparation, including lactoferrin, on Sars-CoV-2 virus particles (Figure 9, Figure 10, Figure 11 and Figure 12).

## 4. Discussion

Novelty and innovation of the work consists of the use of secretions from the nasopharynx on Petri dishes obtained from patients infected with SARS-CoV-2 and their treatment with Salistat SGL03 containing among others lactoferrin. Lactoferrin as a dietary supplement has health-promoting properties that modulate the functioning of the immune system, reflecting anti-bacterial, anti-viral, anti-oxidant, anti-cancer and anti-inflammatory properties. The toxicity to viral particles of Salistat SGL03 containing essential oils, lactoferrin and *L. salivarius* was investigated by analyzing the viability of virus particles in real time [99,100]. It was found that lactoferrin in the preparation may interact with Sars-Cov-2 (COVID-19) virus particles (Figure 9 and Figure 10).

The commercially available preparations on the Polish market with the trade names ProbioticMe, Pharmabest, Optisterin, Lactoferrin, Pharmabest, Jarrow Formulas and Swanson Immuneral are a typical combinations of a probiotic and a prebiotic containing freeze-dried live bacteria: *Lactobacillus bulgaricus*, *Lactobacillus rhamnosus GG*, *Lactobacillus acidophilus* and *Bifidobacterium breve*. Although we cite examples of various preparations with potential anti-viral activity, including the current SARS-Cov2 virus, we do not really know anything about their biological effect on virus particles or bacteria cells. The tested Salistat SGL03 preparation in terms of cytotoxicity to bacteria inhabiting the oral cavity [74] and virus particles, including above analyzed Sars-Cov-2, seems to be the only preparation supporting anti-viral treatment and meeting the expectations of both ordinary people and scientists looking for new natural substances in the pandemic, a probiotic base that, in addition to current vaccines, has ability to help delay the effects of infection with this virus. Salistat SGL03 preparation, widely used in the fight against typically oral bacterial infections by the addition of *L. salivarius* SGL03, lactoferrin, as well as lemon and rosemary oils, has gained a new anti-viral application that can be used in the current pandemic situation. Its universal composition and simple application reduces the level of viral load in oral cavity, which can be observed in the presented experiments.

The presence of lactoferrin itself and its anti-inflammatory properties perfectly harmonize with anti-viral activities. The important role of lactoferrin is to calm down the cytokine storm, which is the main cause of the rapid course of COVID-19 [45,46,47,48,49,50,57,79,101,102,103,104,105,106,107,108,109,110,111,112,113,114,115,116,117,118,119,120,121,122,123,124,125,126,127,128,129,130,131,132,133]. It can be also used to treat osteoporosis because it reduces osteolysis—the destruction of bone cells [134].

The results from this study indicate that there may also be a new indication for this product as an inhibitor of SARS-CoV-2 infection and viral spread. Salistat SGL03 showed anti-viral activity by slowing the multiplication of SARS-CoV-2 in the human nasopharynx (Figure 8 and Figure 9). Currently, in addition to the known preparation ie. Salistat SGL03, newer preparations containing lactoferrin subjected to Lf encapsulation and liposomalization are being tested [101,102]. At present, Lf derivatives against various viruses are being intensively studied in China on children aged 0–10 years [103]. The prevalence of the virus in children 0–10 years after ingesting colostrum from breast milk was found to be only 0.9% [102,103,104]. The course of viral infection in infants was mild and did not require assisted ventilation, and the infection itself rarely evolved into a lower respiratory tract infection [105]. Natural breastfeeding or the extensive use of Lf-containing infant formulas significantly reduces all types of viral infections. In experiments with the poliovirus, it was observed that only lactoferrin, saturated with zinc and not with iron, inhibited viral infection after incubation with cells after virus attachment [106]. This is of particular importance in the case of COVID-19, as zinc supplementation has been proposed as a possible additional intervention in this disease [107]. The use of Lf is very effective in combination with the use of conventional anti-viral drugs in viral diseases against HCV [108] and against SARS-CoV-2 [109,110,111,112,113,114,115]. In the adjuvant treatment of metronidazole in women with recurrent bacterial vaginosis BV, preparations containing a probiotic mixture containing *Lactobacillus acidophilus* GLA-14 and *Lactobacillus rhamnosus* HN001 in combination with bovine lactoferrin were used [116]. It is increasingly recognized that iron overload contributes to the pathogenesis of viral infection [117,118]. Indeed, several of the symptoms of COVID-19, which include inflammation, hypercoagulation, hyperferritinemia and immune dysfunction, are similar to the symptoms of iron overload [117,118]. Iron is highly chemically reactive and potentially toxic due to damage to cellular components, such as lipids (ferroptosis), nucleic acids and cellular proteins, leading to the activation of acute and chronic inflammation. Iron chelators, such as lactoferrin, are generally safe and protect patients from iron overload by exerting immunomodulatory effects by binding to coronavirus receptors, blocking their entry into host cells [118,119]. Literature data show that iron chelators have anti-viral and anti-inflammatory effects [120,121,122,123], which is of high therapeutic value during the current COVID-19 pandemic. Currently, various therapies are used to treat COVID-19 that work on the immune system [124,125]. By activating the immune system, dietary supplements can be used as adjuvants with anti-bacterial and immunomodulating properties to inhibit the spread of the COVID-19 virus [126]. Active substances, which are components of many products include vitamins e.g., vitamin D, probiotics, lactoferrin and zinc, are now extensively clinically tested in patients with COVID-19 respiratory infection [127,128,129,130]. Their molecular effect on viral particles strongly suggests their potential utility in combating COVID-19 [131,132,133]. Earlier literature reports presented by Kucia [74] indicate that the active substances of Salistat SGL03 have an anti-bacterial effect. Preliminary results from the use of a probiotic-based product and its natural ingredients were tested on the bacterial biofilm derived from the oral microbiota of volunteers [2,52,53,54]. In the context of research, they have been tested by people infected with SARS-CoV-2 and are very promising, especially in terms of a new approach to inhibiting and reducing the symptomatic effects of infection caused by this virus. Such preparation can be used in the early prophylaxis of infections, especially by people who may be more susceptible to infections because of a weakened immune function. Therefore, regular use of this preparation during a pandemic may be preceded by adequate stimulation of local and systemic immune responses to inflammation of the throat, nasopharynx and mouth. The use of MIC and MBC tests is the basis of targeted antibiotic therapy and various compounds, such as probiotics or drug susceptibility tests of bacteria or virus particles in chemotherapy. Infections with bacteria that are resistant to antibiotics, such as azithromycin and doxycycline, which are used to treat respiratory infections, are very rare in people infected with the coronavirus. Nowadays, resistance of bacteria and bacteriophages to antibiotics is more common. The lack of homeostasis in the human body can lead to disturbances in the functioning of its basic systems and cause numerous excessive viral or viral-bacterial infections. American political groups from Boston in 2001 published guidelines on avoiding antibiotics in cases of simple cough, colds and viral ulcers [8,29,41,42,43,44,45,46,47,48,49,50,51,52,53,54,55,56,57,58,59,60,61,62,63,64,65,66,67,68,69,70,71,72,73,74,75,76,77,78]. The addition of lactoferrin and its appropriate amount in Salistat SGL03 most likely blocks some viral proteins [74]. Lactoferrin contained in the preparation administered directly into the throat has an antiseptic effect, stimulates the immune system and reduces necrotic TNF-alpha factors in viral infections caused by SARS-CoV-2 and in inhibiting the influenza A/WS/33 virus [97].

## 5. Conclusions

Supplementation with Salistat SGL03, containing lactoferrin and *L. salivarius*, may play an effective protective role, both in preventing viral infection and alleviating the clinical course in infected patients, thereby contributing to the prevention of immune-mediated organ damage [111]. Action of lactoferrin acts on cellular receptors, preventing SARS-CoV-2 virus from anchoring and entering into the cell surface. Further clinical trials on preparations containing lactoferrin are needed. However, the real role of *L. salivarius*, essential oils and lactoferrin in inactivating viral infections in the early stages is still unknown; therefore, further research is needed on cell culture experiments with Vero E6 + SARS lines where different concentrations of Salistat SGL03 that would be added at different time points. Future research should determine the cytotoxic effect and virus concentration—measured in real time, in order to better understand the etiopathogenesis of this disease.

## Figures and Tables

**Figure 1 materials-14-03086-f001:**
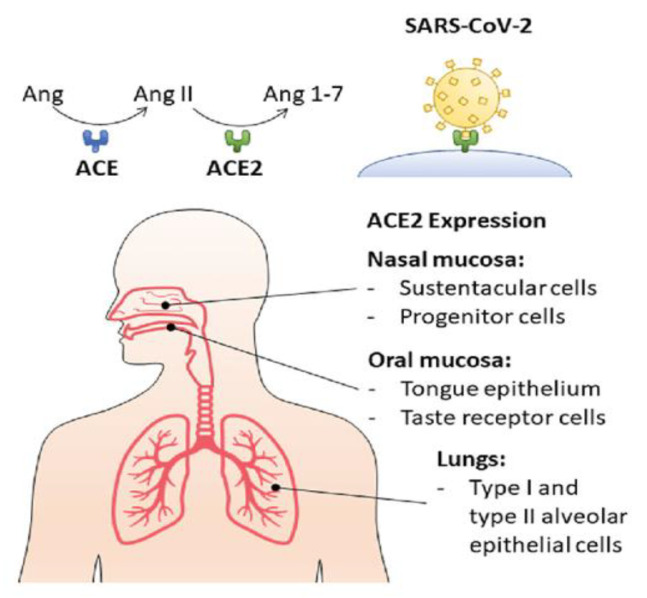
The human oral cavity and nasopharynx to which particles of SARS-CoV-2 attach https://www.researchgate.net/publication/342577883, accessed on 3 June 2021.

**Figure 2 materials-14-03086-f002:**
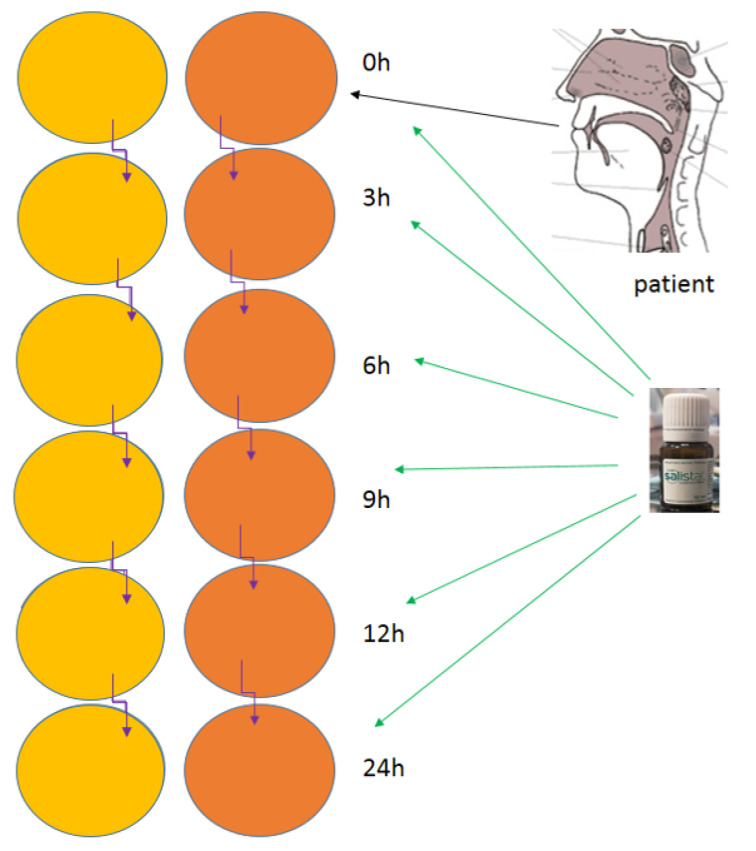
Schematic presentation on experiments part 2.

**Figure 3 materials-14-03086-f003:**
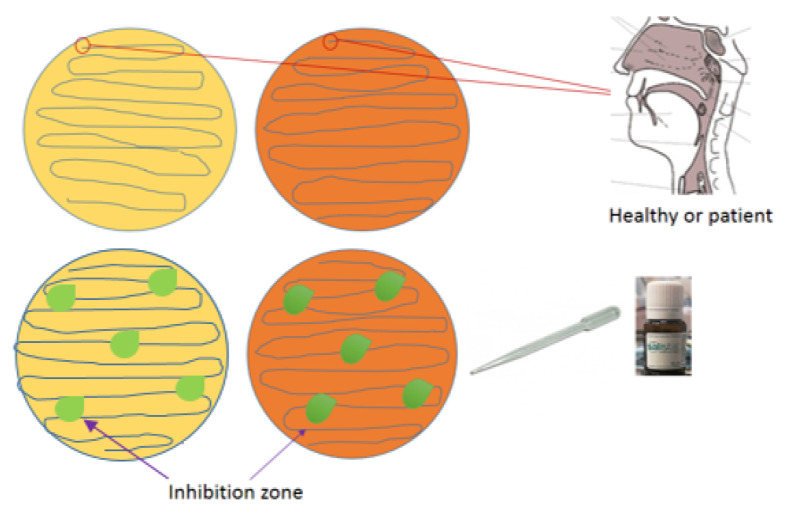
Schematic presentation on experiments part 1 [57,74].

**Figure 4 materials-14-03086-f004:**
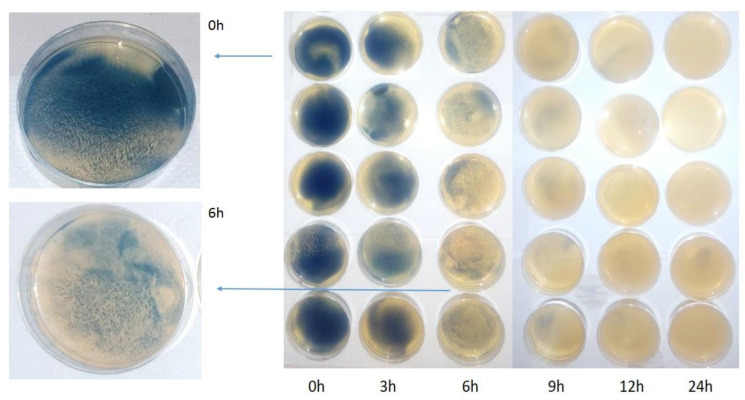
Petri dishes with biological materials of nasopharyngeal secretions from infected virus Sars-CoV-2 patients after 0–24 h incubation treatment with Salistat SGL03 (see Section 2.1).

**Figure 5 materials-14-03086-f005:**
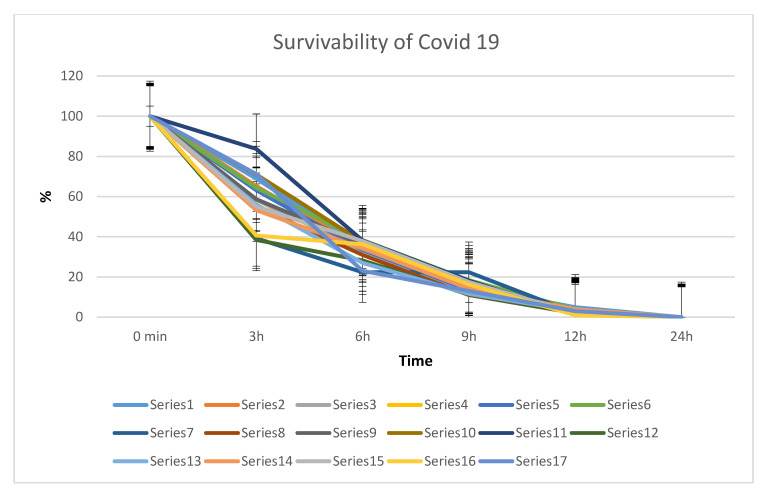
The survivability of SARS-CoV-2 isolated from the patients’ nasopharynx of infected individuals after Salistat SGL03 treatment on specific media plates. Series from 1 to 17 number of individual study participants (infected persons by SARS-CoV-2), (see Section 2.1).

**Figure 6 materials-14-03086-f006:**
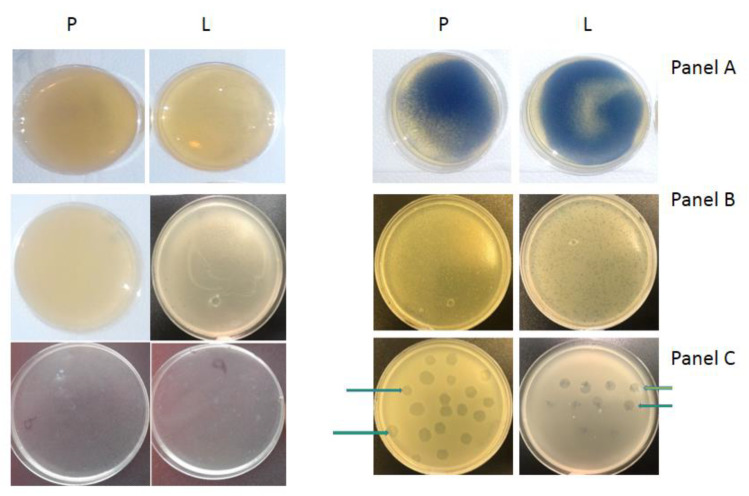
SARS-CoV-2 virus collected from the nasopharynx, after 24 h incubation. Viral inoculum were collected from healthy volunteers and individuals infected by SARS-CoV-2 of the nasopharynx, and were plated on both types of medium complete and selection medium (**A**–**C**).

**Figure 7 materials-14-03086-f007:**
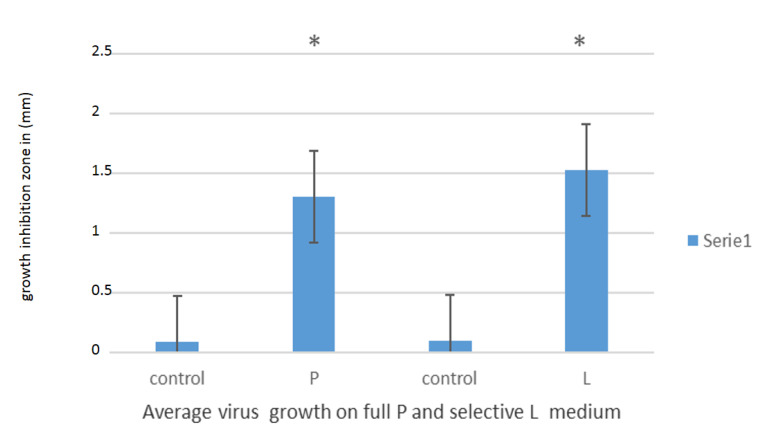
Effect of lactoferrin present in probiotics on inhibition of viral particle growth from people infected with COVID-19 from nasopharynx after 24 h of incubation. Statistical significance at *p* < 0.05 *.

**Figure 8 materials-14-03086-f008:**
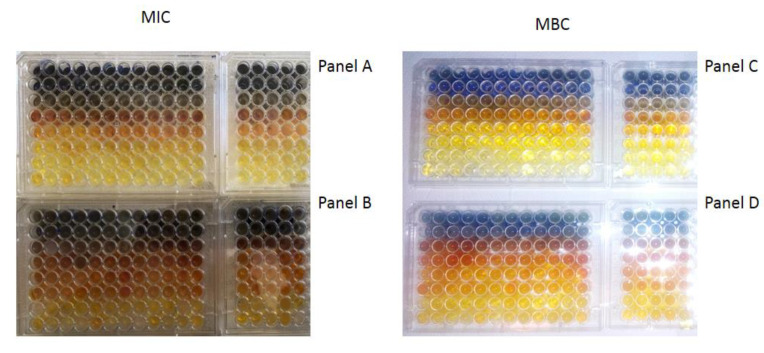
Virus particle analysis using MIC and MBC tests (**A**: healthy volunteers), (**B**: infected individuals with SARS-CoV-2). MBC analysis of tested viral particles (**C**: healthy volunteers) and (**D**: infected individuals with SARS-CoV-2. Lanes from 1 to 17—viral particles with serial dilutions after Salistat SGL03 treatment.

**Figure 9 materials-14-03086-f009:**
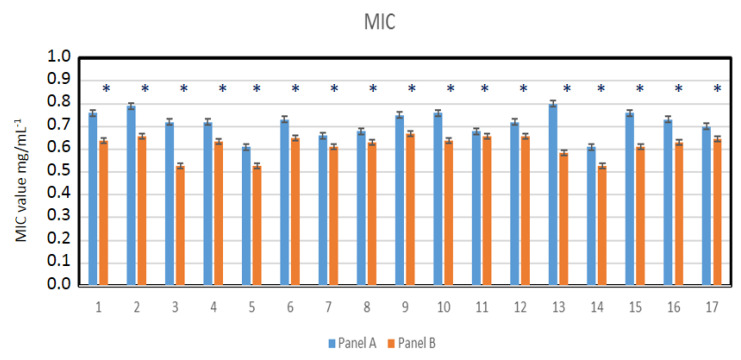
Virus particle analysis using MIC test (panel A: healthy volunteers without SARS-CoV-2) and (panel B: healthy volunteers after treatment of Salistat SGL03). Statistical significance vs. control in all analyzed samples were at * *p* < 0.05. X-axis number of study participants from 1 to 17.

**Figure 10 materials-14-03086-f010:**
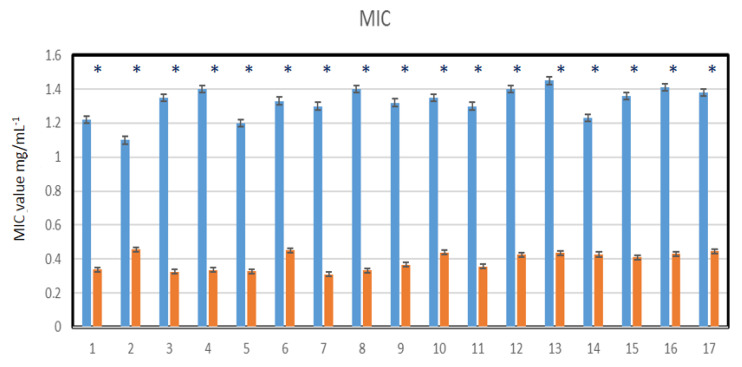
Virus particle analysis using MIC test (panel A- infected volunteers with SARS-CoV-2) and (panel B- infected volunteers after treatment of Salistat SGL03). Statistical significance vs. control in all analyzed samples were at * *p* < 0.05. X-axis number of study participants from 1 to 17.

**Figure 11 materials-14-03086-f011:**
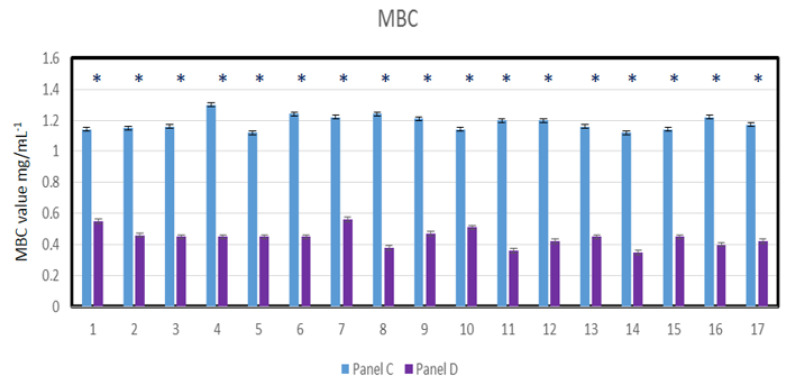
Virus particle analysis using MBC test (panel A- healthy volunteers without SARS-CoV-2) and (panel B- healthy volunteers after treatment of Salistat SGL03). Statistical significance vs. control in all analyzed samples were at * *p* < 0.05. X-axis number of study participants from 1 to 17.

**Figure 12 materials-14-03086-f012:**
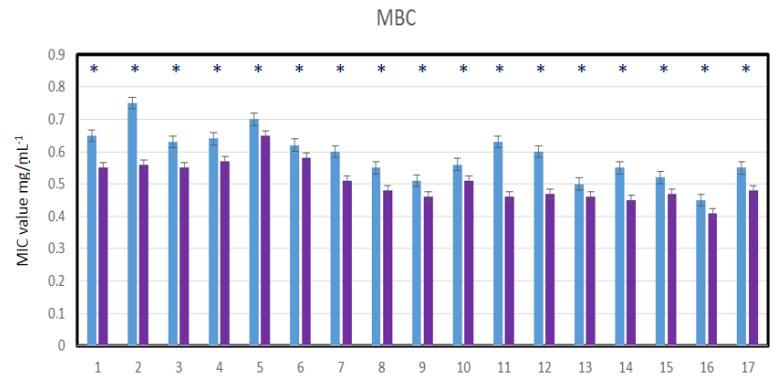
Virus particle analysis using MBC test (panel A- infected volunteers without SARS-CoV-2) and (panel B- infected volunteers after treatment of Salistat SGL03). Statistical significance vs. control in all analyzed samples were at * *p* < 0.05. X-axis number of study participants from 1 to 17.

## Data Availability

On request of those interested.

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
