# Peer review of "Protective Action of L. salivarius SGL03 and Lactoferrin against COVID-19 Infections in Human Nasopharynx"

_materials, 2021, doi:10.3390/ma14113086_

Round 1
Reviewer 1 Report
The concept of the paper gets the attention of the readers. The study is well designed and prosecuted. However, there are few shortcomings.
- The introduction is too lengthy and contains too much irrelevant information causing distraction. I will suggest shorting down the introduction to one and a half pages.
- The methods seem to be complexed kindly divide it into further portions.
- Results are well written however the discussion is not sufficient. Please discuss the results in detail at least 2 pages.
- Overall English is good but try to improve the connections between the sentences.
- The quality of the graphs is not good. I will suggest making the graphs in some sophisticated software that will improve the overall look of the article.
- Please short down the conclusion as well.
- I will suggest to add a graphical abstract.
Author Response
Thank you very much to the reviewer whose extremely valuable comments contributed to the improvement of the quality of our manuscript. All corrections are marked in green.
Reviewer 1
The concept of the paper gets the attention of the readers. The study is well designed and prosecuted. However, there are few shortcomings.
- The introduction is too lengthy and contains too much irrelevant information causing distraction. I will suggest shorting down the introduction to one and a half pages.
The introduction has been shortened to a page and a half as suggested by the reviewer
- The methods seem to be complexed kindly divide it into further portions.
The methods were shortened and divided into smaller fragments as suggested by the reviewer
- Results are well written however the discussion is not sufficient. Please discuss the results in detail at least 2 pages.
the results were discussed as suggested by the reviewer
- Overall English is good but try to improve the connections between the sentences.
We improved the English where it was possible, and we paid attention to connections between sentences
- The quality of the graphs is not good. I will suggest making the graphs in some sophisticated software that will improve the overall look of the article.
The charts based on the data were made in excel. Their quality by enlarging the image was colored in the power point program.
- Please short down the conclusion as well.
The chapter conclusions was arranged with a paragraph added
- I will suggest to add a graphical abstract.
Graphical abstract

Reviewer 2 Report
Highlight changes in yellow in a next revision, please. No track changes.
Consider comments in the entire text.
Dear authors, content similar to the one used in this text can be found in another article from the same authors: 14%, not acceptable, in my perspective.
Also check similar style and figures.
Overall similarity is 24%
Do not use “we”
Do not write like this, and poor English.
An abstract should not contain self-opinions:
“constitute a very promising model in our opinion”
Which “model” d0 authors refer to”. I do not understand the term here…
I am sure authors are aware that ResearchGate links must not be included:
“Figure 1. The human oral cavity and nasopharynx to which belong bacteria of all bacterial 67
complexes [91] https://www.researchgate.net/publication/342577883.”
Remove figures from other unless adapted, and then state so
Do not use the same references continuously:
“[80]. The transmission of SARS-CoV-2 in the air can 103
occur during medical procedures that generate aerosols. In experimental studies, aerosols 104
of infectious samples were generated using high-power jet nebulizers under controlled 105
laboratory conditions [80].”
References are not being presented by number order… Have authors checked the instructions?!
“[1-5].” Then “complexes [91]”
There are many things to be corrected in the text…, typos, spacing, etc, format…
“showed that ,”
“Lactoferrin (another name: lactoferrin [1] LF)”?!
Check proper sacing in units, international unit system… “μg / ml)”
“2.2. Determination of Minimum Inhibition Concentration (MIC) and Minimum Bactericidal 250
Concentration (MBC)” no italics below, does it belong to the heading?
A tiny text does not justify a subsection…
“2.2. Determination of Minimum Inhibition Concentration (MIC) and Minimum Bactericidal 250
Concentration (MBC) 251
All estimations were made using the (MIC) and (MBC) method described previously 252
by Kowalczyk [23,51]. The results were statistically significant at the p < 0.05 level, meas- 253
ured by the Student’s t-test. 254
2.3. Checking if there was SARS-CoV-2 virus in oral microbiota and nasopharynx after treatment 255
of Salistat SGL03 on collected from healthy volunteers and infected individuals”
?!
Captions must be self-explanatory:
“Figure 2. Schematic presentation on experiments part 2.”?!
Why figures as a start in a section?!
Use a table and add all to final list of references:
“Below we present some examples of them: 306
-Act of September 6, 2001, Pharmaceutical Law, Journal of Laws 2001, No. 2001 No. 307
126, item 308
1381 (Announcement of the Marshal of the Sejm of the Republic of Poland of Febru- 309
ary 22, 2019 on 310
the publication of the uniform text of the Act-Pharmaceutical Law, Journal of Laws 311
2019, item 499) 312
-EU Directive 2004/23/EC of the European Parliament and of the Council of 31 March 313
2004 on
setting standards of quality and safety for the donation, procurement, testing, pro- 315
cessing, preservation, storage and distribution of human tissues and cells (OJL102, 7.4. 316
2004, p. 48). 317
Materials 2021, 14, x FOR PEER REVIEW 9 of 23
-Helsinki Declaration of the World Association of Physicians (WMA) Ethical Princi- 318
ples of 319
Conducting Medical Research with Human Participation-October-2013.”
Check italics in parameters… there are are ways to refer to the performed statistics…
“2.4. Statistical Analysis 321
Data are presented as mean ± standard error (SE). Statistica commercial package for 322
Windows (version 5.0) was used for statistical analysis. The assumptions of normality and 323
equal variance were tested on the basis of the Student's t-test. Statistical significance was 324
considered at p <0.05.”
Huge letter in “h”: Figure 4, much bigger than the text…
“series” refre exctly to what?
All axis must have a clear legend, besides the unit in ()… as usual everywhere…
The same in other cases..
“Figure 5. The survivability of SARS-CoV-2 isolated from the nasopharynx of infected indi- 354
viduals infected with the SARS-CoV-2 virus after Salistat SGL03 treatment.”
Why mix 2 different units in the same axis, it makes no sense at all…
Min and h?!
What is panel a and B, see that the reader will look at the figure independently and will not understand again, no clear legends to axis, what is 1 to 17?! When focusing on the figure alone?
The same in all cases: Figures 9 to 12, similar figures should be grouped and detailed captions by letter to each one, after the main caption, should be present.
Here? At the end?
“On the basis of the characteristics of its individual components, we 561
hypothesized that the Salistat SGL03 dietary supplement also showed antiviral activity by 562
slowing down the multiplication of the SARS-CoV-2 in the human nasopharynx (Fig. 563
8 and 9).”
Relating to figures far above…?
Despite the interest of the text, the similarity is bid red flag… unacceptable
Reference 74 is mentioned many times through the text…
Please remove it from conclusions section
Then, the structure of the text needs clear revision, as seen
Conclusions, as abstract, must contain:
Brief contextualization and methodology, main findings and practical implications
I do not feel there is a clear connection between the title, text , the presented abstract and the conclusions, particularly when comparing the last two…
No references from 2021 are available, please update?
The similarity found comprises significant parts of important sections of this text, namely discussion and conclusions, among others
Author Response
Reviewer 2
Thank you very much to the reviewer whose extremely valuable comments contributed to the improvement of the quality of our manuscript. All corrections are marked in green. The names of strains are marked in yellow
Highlight changes in yellow in a next revision, please. No track changes.
Consider comments in the entire text.
Dear authors, content similar to the one used in this text can be found in another article from the same authors: 14%, not acceptable, in my perspective.
Also check similar style and figures.
Overall similarity is 24%
The manuscript has been revised to be similar to our other earlier publications. The introduction to 2 pages has been shortened, materials and methods have been edited. The discussion with conclusions has been corrected. The graphs in the output part of the article have been corrected.

Reviewer 3 Report
Review
Title: Protective action of L. salivarius and lactoferrin against Covid -19 infections in human
nasopharynx
Authors: Marzena Kucia, Ewa Wietrak, Mateusz Szymczak, Michał Majchrzak and Paweł Kowalczyk
In my opinion, this manuscript could be considered for publication, after the following major changes:
- The authors should perfume a thorough spell check of the entire manuscript. Also they should pay attention to using italic font L. salivarius (line 238, 395…………), in vitro (line 340, 387, 404 ..), in vivo (line 372, 404, 435,….…)
- Also please modify sentences like “grown on growth plates” (line 343…)
- In addition please use superscript for 10-6, ml-1 (line 436,………..etc)
- More than that please check carefully the figure legends (there are unnecessary spaces in them )
- Please provide better individual figure for Figure 4. As it is, one cannot conclude anything from those figures.
- Please define Series 1-17 from Figure 5
- The authors should improve the Materials and methods section.
- Please uniform the notation for ml (in text it is both ml and mL)
- Please correct line 532-532, the phrase is interrupted. The same with 552-553
- Moreover, discussions regarding the novelty of the study and comparison between the results in literature and the data obtained by the authors on these materials should also be presented in the manuscript.
Author Response
Reviewer 3
Open Review
Thank you very much to the reviewer whose extremely valuable comments contributed to the improvement of the quality of our manuscript. All corrections are marked in green.
Comments and Suggestions for Authors
Review
Title: Protective action of L. salivarius and lactoferrin against Covid -19 infections in human
nasopharynx
Authors: Marzena Kucia, Ewa Wietrak, Mateusz Szymczak, Michał Majchrzak and Paweł Kowalczyk
In my opinion, this manuscript could be considered for publication, after the following major changes:
- The authors should perfume a thorough spell check of the entire manuscript. Also they should pay attention to using italic font L. salivarius (line 238, 395…………), in vitro (line 340, 387, 404 ..), in vivo (line 372, 404, 435,….…)
All suggestions of the reviewer have been taken into account, strain names like eg. L. salivarius have been corrected in italics (lines 238, 395 …………), in vitro (line 340, 387, 404 ..), in vivo (line 372, 404, 435,….…
- Also please modify sentences like “grown on growth plates” (line 343…)
The the sentence has been modified
- In addition please use superscript for 10-6, ml-1 (line 436,………..etc)
has been revised everywhere in the text
- More than that please check carefully the figure legends (there are unnecessary spaces in them )
the legends in the figures are described in more detail and spaces have been removed
- Please provide better individual figure for Figure 4. As it is, one cannot conclude anything from those figures.
figure 4 has been enlarged and widened to better show nasopharyngeal discharge in the dishes
- Please define Series 1-17 from Figure 5
The series 1-17 in figure 5 are defined under the figure description
- The authors should improve the Materials and methods section.
the materials and methods section has been improved
- Please uniform the notation for ml (in text it is both ml and mL)
the provision has been unified
- Please correct line 532-532, the phrase is interrupted. The same with 552-553
lines 532-532, 552-553 with phrases that were broken have been corrected in the chapter Literature
- Moreover, discussions regarding the novelty of the study and comparison between the results in literature and the data obtained by the authors on these materials should also be presented in the manuscript.
The article presents a discussion on the novelty of research and the comparison of the results from the literature data with the data obtained by the authors on these materials.

Round 2
Reviewer 1 Report
The abstract has not been reduced according to previous instructions. Moreover, the intorduction further need to be compressed to one and half pages.This article in the pesent form is damaging the whole scientific report.
Author Response
The abstract has been reduced as per previous instructions. The introduction has been compressed to one and a half pages by Editorial template. The figures have been corrected with the captions under them. English was also checked in terms of form and content. All corrections are marked in yellow.

Reviewer 2 Report
Highlight changes in yellow in a next revision, please. No track changes.
Consider comments in the entire text.
The authors have not answered to all my comments in detail.
I will be consistent with my previous review report.
Nothing really changed
The discussion has no implications in the novelty of the manuscript
similarity remains in 24%
16% alone from one... source, not acceptable
Author Response
The discussion has been changed and the probability has been lowered with our previous manuscripts. All changes have been taken into account from the previous review and marked in yellow.

Reviewer 3 Report
The authors have made some modifications as per my suggestions but the manuscript still needs some improvements before being considered for publication:
- Fig 4 is still at a poor resolution. The authors did not provide other adequate figures, they just enlarged the ones that were presented initially. They are still at a poor resolution and no scientific conclusion could be draw from here.
- I just realized that Fig 9 to 12 have no axis. This is unacceptable for a scientific paper.
- There are still a lot typos, and some other English grammar mistakes in the manuscript.
- Series 1-17 were not described in the figures.
Author Response
- Fig. 4 we showed another set of plates from infected patients, we improved the drawing and its resolution.
- Fig. 9 Up to May 12 already assigned axes
- In the manuscript, we tried to correct all typos and some other grammatical errors in the English language.
- Series 1-17 are described under the figures.
All corrections are marked in yellow.

Round 3
Reviewer 2 Report
Consider comments in the entire text.
Do not use “our” or personal pronouns
The authors differentiated the text in order to move away from similarity but the plagiarism is present.
Even headings are the same as previous works
Same figures and e even captions
To change the discussion section lowered the similarity, still present in any case
I will be consistent with my previous review report.
The discussion has no implications in the novelty of the manuscript
Author Response
Reviewer 2
Once again, thank you very much for the all suggestions of the reviewer, which contributed to the improvement of the substantive quality of our manuscript.
Comments and Suggestions for Authors Consider comments in the entire text.
Do not use “our” or personal pronouns
Pronouns "our" and personal pronouns were removed from the manuscript as suggested by the reviewer
Even headings are the same as previous works
The headline part in the descriptions has been changed
Same figures and e even captions
The numbers are the same because we have worked with raw data and only such data is available to us. We presented the results as is because we thought they would be more legible. With the approval of other reviewers. The results are not identical to other works, they are an individual new part of the experience that we present in our work
The discussion has no implications in the novelty of the manuscript
The discussion has been re-arranged in relation to the description of Salistat SGL03. Currently, there are no reports in the literature about other lost preparations that would have such a unique composition and formula and would act in two ways on both bacterial cells in the oral cavity {74] and on the particles of the Covid-19 virus (a new property of Salistat - discovered by our team). Therefore, in the discussion, we focused more on the description of the action of the individual components of the preparation in terms of their new biological effects in antiviral activity. We did not describe other studies because they do not exist, so we could not refer to them.
To change the discussion section lowered the similarity, still present in any case
The authors differentiated the text in order to move away from similarity but the plagiarism is present.
We improved as much as we could (without having an anti-plagiarism program) the similarities of sentences with our previous works or works on related topics so as to avoid being accused of plagiarism or self-plagiarism. It is known that there is a certain body of formulations in science which cannot be avoided in order to present a similar research problem
Captions under the figures have been re-arranged and corrected
A novelty in the manuscript is the new antiviral activity of Salistat SGL03 in addition to the previously defined antimicrobial activity described in the work cited in reference 74 in this manuscript
We tried to correct the manuscript as best we could, following all reviewers' recommendations.
We kindly ask the current reviewer, despite his many critical comments, to agree and enable the publication of our article. We believe that this publication (reaching a wide range of readers) will help many people to better protect themselves from infections due to Covid-19 infection and reduce the risk of new fatal infections after using this preparation.

Reviewer 3 Report
The authors have addressed all my comments.
Author Response
Once again, thank you very much for the all suggestions of the reviewer, which contributed to the improvement of the substantive quality of our manuscript.
Captions under the figures have been re-arranged and corrected
A novelty in the manuscript is the new antiviral activity of Salistat SGL03 in addition to the previously defined antimicrobial activity described in the work cited in reference [74] in this manuscript
The discussion has been re-arranged in relation to the description of Salistat SGL03.
